# Polyamine Transport Protein PotD Protects Mice against *Haemophilus parasuis* and Elevates the Secretion of Pro-Inflammatory Cytokines of Macrophage via JNK–MAPK and NF–κB Signal Pathways through TLR4

**DOI:** 10.3390/vaccines7040216

**Published:** 2019-12-14

**Authors:** Ke Dai, Xiaoyu Ma, Zhen Yang, Yung-Fu Chang, Sanjie Cao, Qin Zhao, Xiaobo Huang, Rui Wu, Yong Huang, Qigui Yan, Xinfeng Han, Xiaoping Ma, Xintian Wen, Yiping Wen

**Affiliations:** 1College of Veterinary Medicine, Sichuan Agricultural University, Chengdu 611130, China; daike1990@stu.sicau.edu.cn (K.D.); xiaoyu@stu.sicau.edu.cn (X.M.); yzhen@stu.sicau.edu.cn (Z.Y.); csanjie@sicau.edu.cn (S.C.); zhao.qin@sicau.edu.cn (Q.Z.); huangxiaobo@sicau.edu.cn (X.H.); wurui1977@sicau.edu.cn (R.W.); hyong601@sicau.edu.cn (Y.H.); yqg@sicau.edu.cn (Q.Y.); hanxinf@sicau.edu.cn (X.H.); mxp886@sicau.edu.cn (X.M.); xintian@sicau.edu.cn (X.W.); 2Department of Population Medicine and Diagnostic Sciences, College of Veterinary Medicine, Cornell University, New York, NY 14850, USA

**Keywords:** *Haemophilus parasuis*, polyamine, PotD, pro-inflammatory cytokines, signal pathway

## Abstract

The *potD* gene, belonging to the well-conserved ABC (ATP-binding cassette) transport system *potABCD*, encodes the bacterial substrate-binding subunit of the polyamine transport system. In this study, we found PotD in *Haemophilus* (*Glaesserella*) *parasuis* could actively stimulate both humoral immune and cellular immune responses and elevate lymphocyte proliferation, thus eliciting a Th1-type immune response in a murine immunity and infection model. Stimulation of Raw 264.7 macrophages with PotD validated that Toll-like receptor 4, rather than 2, participated in the positive transcription and expression of pro-inflammatory cytokines IL–1β, IL–6, and TNF–α using qPCR and ELISA. Blocking signal-regulated JNK–MAPK and RelA(p65) pathways significantly decreased PotD-induced pro-inflammatory cytokine production. Overall, we conclude that vaccination of PotD could induce both humoral and cellular immune responses and provide immunoprotection against *H. parasuis* challenge. The data also suggest that *Glaesserella* PotD is a novel pro-inflammatory mediator and induces TLR4-dependent pro-inflammatory activity in Raw 264.7 macrophages through JNK–MAPK and RelA(p65) pathways.

## 1. Introduction

*Haemophilus* (*Glaesserella*) *parasuis* is a pleomorphic, nonmotile, nicotinamide adenine dinucleotide (NAD)-dependent bacterium belonging to the family Pasteurellaceae and causes Glasser’s disease (GD) in piglets [1]. *H. parasuis* is primarily readily isolated in weaner pigs, followed by finisher pigs and sows [2]. It normally colonizes the upper respiratory tract of swine and includes strains with diverse genetics and pathogenicities [3]. When a host is under stress, virulent strains can penetrate the mucosal barrier and enter the bloodstream, causing severe vascular lesions and multiple syndromes [4]. Clinical signs of GD are characterized by pneumonia, meningitis, polyarthritis, and fibrinous polyserositis, giving rise to significant economic losses in the pork industry throughout the world [5,6]. So far, over 15 serotypes and at least 15 antimicrobial resistance genes (14 plus another *tetO* gene) have been found in this species [7,8,9]. As a result, the current regimen of curing this disease relies heavily on antibiotic therapy, and it is difficult to prevent this disease by immunization of inactivated vaccines due to the lack of crossprotection between different serotypes [4]. It is urgent to determine the mechanism of its pathogenesis in depth, and to find an effective protective vaccine [10].

Polyamines, mostly consisting of putrescine, spermidine, and spermine, are natural polycationic biomolecules which exist in all living organisms [11]. They were discovered when Antonie van Leeuwenhoek isolated some visible ‘three-sided’ crystals from human semen using his microscope in 1678. Afterwards, the empirical formula of the crystals was deduced in 1924 [12,13]. Polyamines participate in multiple physiological functions. They are essential for cell growth and cell death processes, such as apoptosis [14]. Among the three substances, spermidine has remarkably neuroprotective and cardioprotective effects and can actively stimulate anticancer immunosurveillance in rodent models [15]. In microorganisms, polyamines are thought to be actively transported from periplasm to cytoplasm via a well-conserved ABC (ATP-binding cassette) transport system *potABCD* [16]. This gene cluster sharing a common operon encodes a functionally highly relevant and specialized high-affinity transport system *PotABCD*, wherein PotD is a spermidine/putrescine-binding periplasmic protein, serving as the direct binding receptor for both spermidine uptake and excretion, while PotA protein—an ATPase—binds ATP for spermidine uptake in conjunction with transport channels formed by PotB and PotC, which anchor on the inner membrane [17,18,19]. It has been demonstrated that PotD preferentially binds spermidine over putrescine using surface plasmon resonance (SPR) and radioisotope-labeling techniques in *Synechocystis* sp. and *Escherichia coli* [14,20]. Intracellular polyamine levels are tightly regulated in their biosynthesis, degradation, and transport through a sophisticated mechanism [12,21,22]. Five conserved polyamine-binding residues (Asp–Glu–Trp–Trp–Asp) have been found in the PotD amino acid sequence [23]. This is usually adopted to identify its polyamine-binding capacity in different PotD protein in various species. 

Thus far, besides polyamine uptake and excretion, other biological functional studies demonstrated that PotD is linked to the stimulation of the expression of SOS response-related genes and biofilm formation in *E. coli* [24]. In *Streptococcus pneumoniae*, PotD is involved in pathogenesis within various infection models [25,26], and immunization with PotD induces a vigorous antibody response and, thus, can provide strong immune-protective efficacy in a mouse model [27,28]. In *Legionella pneumophila*, more puzzlingly complex phenotypes were observed when *potD* gene was knocked out, such as defects in the abilities to form filaments, tolerate high Na^+^ concentrations, associate with macrophages and amoeba, recruit host vesicles to the *Legionella*-containing vacuole (LCV), and to initiate intracellular growth [29]. However, the link between PotD and polyamines in terms of biological functions still remains unclear in *H. parasuis*. In previous studies, we and others have used 2-dimensional gel electrophoresis (2-DE) and proteomic analysis to show that PotD is a stress-responsive protein and one of the differentially expressed virulence-related proteins [30,31]. However, little evidence of the exact biological function of PotD has been unveiled in *H. parasuis*. 

Eukaryotic cells have a fine-tuned immune recognition system against extracellular stimulus signaling molecules. Ample evidence exists to prove that microbial sensors on cell surface utilize pattern recognition receptors (PRRs) that recognize pathogen-associated molecular patterns (PAMPs) or, alternatively, newly defined microbial-associated molecular patterns (MAMPS) [32] from different microorganisms, thus tightly regulating specific mechanical stress-mediated activation of signal pathway(s) that may lead to inflammation, apoptosis, and atherosclerosis. PAMPs/MAMPs range from microbial glycan, bacterial glycolipids, flagellin to viral and bacterial nucleic acids [33]. As counterparts, the most common PRRs include Toll-like receptor (TLRs, receptors of innate immunity) [34], NOD-like receptor (NLRs) [35,36], RIG-like receptor (RLR) [37], AIM2-like receptors (ALRs), MDA5 (melanoma differentiation associated gene 5), C-type lectin family, scavenger receptor (SR), and mannose receptor (MR) [38]. It has been well-established that TLRs and other PRRs occupy a vital position in the mammalian innate/natural immunity. Interaction of TLRs with their ligands promotes the transcription of genes encoding inflammatory cytokines such that signaling by certain TLRs has an important function in protection against both foreign pathogens and endogenous inflammatory ligands [39]. In *H. parasuis*, Siglecs (Sialic acid-binding Ig-like lectins) is another receptor constituting the inhibitory mechanism against an exacerbated innate response initiated by PRRs that could be dangerous to the host [40]. Additionally, CD163 (M130/p155) has also been reported to be a cell surface receptor of peripheral blood mononuclear cells (PBMCs) from pigs, and it plays crucial roles in immune response and bacterial sepsis caused by *H. parasuis* [41].

In this research, we focus on the immunoprotection of PotD in *H. parasuis* (HpPotD) along with the water adjuvant MONTANIDE™ GEL 01. First, we tested the immunoprotective effects of recombinant PotD (rPotD) in mice, then we analyzed the MAPK and NF–κB signal pathways induced by rPotD on a mouse monocyte/macrophage leukemia cell line Raw 264.7, and its participating TLRs. The aim of this study is to determine the mechanism of the induction of innate immunity within murine models and the details of inflammatory response to Hp–rPotD protein exposure.

## 2. Materials and Methods

### 2.1. Animals and Ethics Statement

Adult female specific pathogen-free (SPF) BALB/c mice weighing 20–25 g (6–8 weeks old) were obtained from Dossy Experimental Animal Co., Ltd, Chengdu, China. All animal protocols were strictly performed under the approval from the Institutional Animal Care and Use Committee of Sichuan Agricultural University, Sichuan, China (Approval Number IACUC#RW2016-090). The China Regulations for the Administration of Affairs Concerning Experimental Animals (1988), and the Guide for the Care and Use of Laboratory Animals of the Ministry of Science and Technology of the People’s Republic of China were followed. Our experimental protocols were designed to provide comfort and minimal stress for our experimental mice.

### 2.2. Strains, Plasmids, Primers and Bacterial Growth Conditions

Primers used in this study are listed in Table 1. *Escherichia coli* DH5α (Biomed, Beijing, China) and BL21(DE3) (Biomed, China) were cultured in liquid Luria-Bertani (LB, BD-Difco, Franklin Lakes, NJ, USA) medium or on LB agar (Invitrogen, Shanghai, China) plate. Wild-type *H. parasuis* SC1401 (highly transformable, serotype 11) and SH0165 (highly virulent, serotype 5) were grown in tryptic soy broth (TSB, Difco, USA) or on a tryptic soy agar (TSA, Difco, USA) plate supplemented with 5% inactivated bovine serum (Solarbio, Beijing, China) and 0.1% (w/v) nicotinamide adenine dinucleotide (NAD, Sigma-Aldrich, St. Louis, MO, USA) (TSB++ and TSA++). When necessary, the media were supplemented with 50 µL kanamycin (Kan, 100 mg/mL), 100 µL ampicillin (100 mg/mL). All strains were shaken at 230 r/min (1 r = 2πrad) at 37 °C. 

Lowercase letters are homologous recombination fragments required for in-fusion clone to the vectors. Restriction enzyme sites *Bam*HI (ggatcc) and *Hin*dIII (aagctt) are shown in bold characters.

### 2.3. Expression/Purification of Recombinant PotD Protein, and Removal of Endotoxin Contamination

Expression of recombinant protein of PotD (rPotD) was performed using an *E. coli* expression system. Briefly, PCR fragments containing *H. parasuis potD* gene minus a 22 amino acid signal peptide sequence were amplified from genomic DNA of SC1401 using primers P1 + P2 (*potD*-pET-F/R, Table 1). The resulting 1015 bp PCR products were cloned into the restriction enzyme sites *Bam*HⅠ and *Hin*dⅢ of linearized pET-28a (+) (TaKaRa) using ClonExpress Ⅱ One Step Cloning Kit (Vazyme, Jiangsu, China), giving rise to pET-*potD*. When reaching an optical density at 600 nm (OD_600nm_) of 0.5 to 0.6, *E. coli* BL21(DE3) bearing pET-*potD* was induced to express by the addition of 1 mM isopropyl-β-D-thiogalactopyranoside (IPTG) for 5 h. The culture was spun at 5000*g* for 10 min, resuspended in 50 mM Tris-HCl, and then lysed by sonication. The lysate was further centrifuged at 12,000*g* for 10 min to pellet cell debris. The His-tag fusion recombinant protein was purified by Ni affinity chromatography using a Bio-Scale^TM^ Mini Nuvia^TM^ IMAC Ni-Charged Resin (5 mL prepacked column, Bio-Rad, CA, USA). Purified rPotD was dialyzed with 2 L of PBS for 2 days and then analyzed by SDS-PAGE electrophoresis. After being concentrated by a 10 kDa ultrafiltration device (Millipore, Bedford, MA, USA), concentration of recombinant protein was determined using a Modified BCA Protein Assay Kit (Sangon Biotech, Shanghai, China). A ToxinEraser^TM^ Endotoxin Removal Resin (Genscript, Pescataway, NY, USA) was used to treat the purified protein for bringing down endotoxin contamination to lower than 0.1 EU/mL [42]. Chromogenic endpoint LAL assay was adopted to quantify the endotoxin as previously described [43]. 

### 2.4. Immunization

Mouse immunization was carried out to determine immune responses and protection against challenging from a heterogeneous *H. parasuis* SH0165. We performed both active and passive immunization analyses.

For active immunization, thirty SPF BALB/c 6–8-week-old mice were allocated randomly into rPotD, adjuvant, and rPotD+adjuvant groups (T3–T5, 10 mice per group), and another 20 mice were allocated to mock (T1) and PBS (T2) control groups. Each mouse in T5 was routinely immunized with 0.1 mg of rPotD plus one-tenth volume of adjuvant (total volume of 0.2 mL) subcutaneously on the backside. Mice in T3 was vaccinated with only rPotD without adjuvant. In parallel, hosts in groups T2 or T4 were injected with equivalent volume of PBS or adjuvant solely, respectively, as negative controls. The mouse in the mock groups T1 received no treatment, serving as a blank control. All mice were vaccinated and maintained on the same program in the same breeding environment with unlimited food and water. Blood samples were collected two days before the challenge for passive immunization analysis usage. The surviving mice were humanely euthanized 7 days post-infection (dpi).

### 2.5. Analysis of Cellular Immune Response of Mice Using Immunofluorescent (IF) Assay and Flow Cytometry

After mice had been immunized with rPotD protein three times, three animals each from the mock group (immunized with PBS/adjuvant) and immunized group (immunized with rPotD+adjuvant) were humanely euthanized, with their spleens being treated with FFPE (formalin/paraformaldehyde-fixed, paraffin-embedded) and used for IF staining to determine the elevation of immunocyte CD3^+^/CD4^+^/CD8^+^ (CD positive T-cells). Photos were shot using a fluorescence microscope (Nikon Eclipse ci&NIKON DS-U3) to detect cyanine (CY3)-labeled fluorescent secondary antibodies (CY3, EX WL: 510–560 nm, EM WL: 590 nm) and analyzed by an immunofluorescence assay using the CaseViewer software. While the fluorescence signal intensities were shown as integrated optical density (IOD) values, analyzed by ImageProPlus software. Relative values of IOD_R_ were calculated by the ratio of the IOD of CY3-fluorescence intensity (IOD_C_) and DAPI (IOD_D_).

The blood of the three groups was applied to flow cytometry (FCM) analysis to measure lymphocyte subtype. Briefly, 100 μL portions of heparin anticoagulant peripheral blood cells were incubated with 10 μL hamster anti-mouse CD3-FITC, CD4-Percp, and CD8-PE (BD Biosciences, San Diego, CA, USA) for 30 min at RT in the dark. Thereafter, RBC lysing solution (Invitrogen, Carlsbad, CA, USA) was added and incubated for another 5 min. Subsequently, the cocktail was centrifuged at 300*g* for 5 min, and the supernatant was discarded. Cells were resuspended in 0.5 mL PBS (pH = 7.2–7.4) and detected by a flow cytometer (FACSVerse, BD Biosciences, San Jose, CA, USA) within 1 h. All experiments were performed in triplicate with at least 10,000 cell counts per treatment. The frequencies of T-helper cells (CD3^+^CD4^+^) and cytotoxic T-cells (CD3^+^CD8^+^) were analyzed by Flowjo software (Flowjo LLC, Ashland, OR, USA). The experiments were independently performed at least three times in triplicate.

### 2.6. Analysis of Lymphocyte Proliferation and Detection of Cytokines Using ELISA

Lymphocyte proliferation assays were performed as previously described with some modifications [44]. Two weeks after booster immunizations of rPotD plus MONTANIDE™ GEL 01 adjuvant, primary spleen lymphocytes from mice in both the mock and immunized groups were separated and maintained for 72 h at 37 °C with 5% CO_2_. Briefly, mice were humanely euthanized and their spleens were isolated aseptically and processed by gentle mechanical disruption using a 10 mL disposable syringe and a sterile cell strainer (40 μm in aperture, Gibco, Grand Island, NY, USA). Splenocytes were flushed down by 5 mL incomplete RPMI-1640 medium (Gibco, USA) suspended in Hank’s balanced salt mixture (HBSS, Solarbio, Beijing, China). Red Blood Cell Lysis Buffer (ACK lysis buffer, Coolaber, Beijng, China) was added to remove erythrocytes, and cell suspensions were centrifuged for 5 min at 300*g*. The erythrocyte-free cells were washed three times with HBSS and then resuspended in complete RPMI-1640 medium. The number of cells was counted and adjusted to 10^7^/mL, and 200 μL of cells were added to 96-well culture plates (Costar, Corning Inc., Corning, NY, USA). The cells were stimulated with the rPotD proteins (10 μg/well), the same volume of concanavalin A (ConA, final mass of 50 μg/well, Sigma-Aldric, St. Louis, MO, USA) or RPMI-1640 medium (mock group), and incubated. Lymphocyte proliferation assays were performed using the MTS reagent with a cell proliferation kit (CellTiter 96^®^ AQueous One Solution Reagent, Promega Inc., Madison, WI, USA) according to the manufacturer’s specifications. The lymphocytes were incubated in 96-well culture plates with MTS reagent for 4 h, and the absorbance was measured at 490 nm (A_490nm_) using a Varioskan Flash (Thermo Scientific, Waltham, MA, USA). 

To determine the type of immune response, we tested the expression of cytokines. The supernatants from cell cultures were harvested after 72 h and stored at −80 °C. The levels of IL-1β, IL-4, and IFN-γ were determined in culture supernatants using the double-antibody sandwich ELISA kits (Beijing 4A Biotech Co., Ltd., Beijing, China), as per manufacturer’s specifications.

### 2.7. Mouse Challenge

A week after the booster immunizations (on the 28th day), all mice were challenged intraperitoneally with a lethal dose of highly virulent wild-type *H. parasuis* strain SH0165 (1.4 × 10^9^ CFU, 0.5 mL), except for the mice in the mock group. All mice were monitored continuously daily after the challenge for the presence and severity of respiratory symptoms and general illness or mortality. Clinical symptoms were scored according to murine activity scores, which were recorded by the sum of the following grading standards: poor spirit (1 point), loss of appetite (1 point), hunched posture (1 point), increased heart rate (1 point), fur loss (1 point), and weight loss (1 points). Mice showing more severe symptoms received higher scores [45]. After mice has been challenged for 72 h, three animals in each group were humanely euthanized, their hearts, livers, spleens and lungs being necropsied and treated with FFPE and used for histological examination (H&E) analysis to determine the pathological lesion before or after immunization. To explore the protectivity elicited by mouse IgG antibodies fraction against PotD, a further passive immunization and challenge was analyzed. Briefly, another five groups of 6–8-week-old mice (10 mice per group, T1’–T5’, corresponding to T1–T5) were intraperitoneally injected with 100 μL of pooled serum collected from the immunized mice 2 h before the challenge with a lethal dose of wild-type *H. parasuis* SH0165. All mice were monitored continuously daily and their clinical symptoms were determined routinely as above. The surviving mice were humanely euthanized 7 dpi. The experiments were independently performed two times.

### 2.8. Bacterial Clearance

In order to quantify bacterial clearance after immunization, thirty SPF BALB/c 6–8-week-old mice were allocated randomly into rPotD, adjuvant, and rPotD+adjuvant groups (6 mice per group), and another 12 mice were equally and randomly allocated to mock and PBS control groups. The same grouping scheme also apply to the passive immunization groups. These mice were immunized with rPotD, adjuvant, rPotD+adjuvant, nothing, or PBS (active immunization group), or pre-injected with their corresponding serum collected from the immunized mice as above (passive immunization group). All mice were challenged intraperitoneally with a half lethal dose (HLD) of *H. parasuis* strain SH0165 (2.4 × 10^8^ CFU, 0.5 mL), except for the mice in the mock group. Twelve hours after the booster immunization, serial dilutions were prepared for equivalent volume of lung homogenates from PBS/PotD alone/adjuvant and PotD+adjuvant-inoculated mice in active and passive immunization groups (10^1^–10^7^ dilutions in sterile PBS; four lung tissue samples in each immunization group). One hundred microliters per diluted sample were plated on TSA++ supplemented with 200 µL nalidixic acid (100 mg/mL). Plates were incubated at 37 °C overnight, and visible bacterial colonies were counted manually. The number of visible bacteria in the mock group was used as a background reference value. Colonies were identified by PCR. The surviving mice were humanely euthanized immediately after this experiment, so as not to cause them pain.

### 2.9. Cell Line and Culture Conditions

In the next studies, we adopted a mouse monocyte/macrophage leukemia cell line RAW 264.7 to dissect immune indices and signal pathways. RAW 264.7 was cultured in complete modified RMPI-1640 medium (Gibco, USA) containing 10% fetal bovine serum (Gibco, USA), and maintained at 37 °C a humidified incubator with 5% CO_2_. Macrophages (1 mL) were added to 12-well cell culture plates at 1 × 10^6^ cells per well. For simulation of PotD protein challenge in vitro, RAW 264.7 cells were stimulated by rPotD protein at a final concentration of 15 or 30 μg/mL for concentration-dependent tests. For detection of immune responsive cell receptors and signal pathways, the cells were collected at 12 h after stimulation. To reduce the possible contamination by bacterial lipopolysaccharide (LPS), polymyxin B (PMB) was added at a final concentration of 10 µg/mL throughout the experiments, except where LPS was utilized as a positive control.

### 2.10. Detected of Pro-Inflammatory Cytokines Induced by PotD Using ELISA and Quantitative Real-Time PCR

For ELISA detection, rPotD was added to Raw 264.7 macrophage culture medium at a final concentration of 15 or 30 μg/mL. LPS (Solarbio, without polymyxin B) was added to a group of three wells in parallel at a concentration of 200 ng/mL to serve as a positive control. An equivalent volume of PBS was added to the negative control group. The cultures were incubated for 12 h, then spun down. The supernatants from cell cultures were harvested after 72 h and stored at −80 °C. The levels of IL-1β, IL-6, and TNF-α were determined in culture supernatants using the double-antibody sandwich ELISA kits (Beijing 4A Biotech Co., Ltd., China), as per the manufacturer’s specifications. The experiments were independently performed at least three times in triplicate.

For quantitative real-time PCR (qRT-PCR), all treated cultures above were set to incubate for 12 h. Total RNA in each group was extracted using a TRIzol Kit (TaKaRa, Shiga, Japan). Their concentrations were examined by measuring the absorbance at 260 nm (A_260nm_) using a spectrophotometry Smartspec^TM^ Plus (Bio-Rad). The purity of RNA was determined by the ratio of absorbance at 260 and 280 nm. Integrity of RNA was verified using agarose gel electrophoresis (AGE). RT-PCR was performed using an iScript cDNA Synthesis Kit (Bio-Rad). qRT-PCR was performed by using an iTaq^TM^ universal SYBR Green Supermix (Bio-Rad) in a Lightcycler96 (Roche, Basel, Switzerland) system. The relative quantification of 2(−ΔΔC(T)) method was used to test transcriptional levels of cytokines IL-1β, IL-6, and TNF-α. Primers for qPCR are listed in Table 1. GAPDH was used as internal reference gene. The experiments were independently performed at least three times in triplicate.

### 2.11. Toll-Like Receptors 2 (TLR2) and 4 (TLR4) Blocking Assay

To determine macrophage recognition receptors of PotD, we blocked the corresponding Toll-like receptors 2 (TLR2) and 4 (TLR4), and both TLR2/4 on Raw 264.7 macrophage using their corresponding monoclonal antibodies (mAb) (Ultra-LEAF^TM^ Purified anti-mouse/human CD282 (TLR2) and LEAF^TM^ Purified anti-mouse TLR4 (CD284)/MD2 Complex, BioLegend, San Diego, CA, USA). Both antibodies were added to a final concentration of 10 µg/mL. rPotD protein was added to cultured macrophages to a final concentration of 15 µg/mL for 12 h after 30 min pre-treatment with two blocking antibodies. Raw 264.7 were spun down and supernatant was collected for routine detection of cytokines IL-1β, IL-6, and TNF-α using double-antibody sandwich ELISA kits (Beijing 4A Biotech Co., Ltd.). Results were analyzed to determine cytokine response levels and to confirm identity of receptors contributing to the response to rPotD exposure.

### 2.12. Western Blotting Assay for Induction of Pro-Inflammatory Cytokines by PotD

Raw 264.7 macrophage was treated with 15 μg/mL of purified rPotD for 12 h, then spun down. The total protein of rPotD-treated and untreated macrophages was extracted. Briefly, the pelleted suspension cells were resuspended in ice-cold PBS (pH = 7.2−7.4) for 15 min, supplemented with Halt Protease Inhibitor Cocktail and Halt^TM^ Phosphatase Inhibitor Cocktail (Thermo Scientific) to prevent proteolysis and maintain phosphorylation status of proteins. Afterwards, the collected cells were resuspended in phosphate buffered saline (Roche) and RIPA lysis buffer and allowed to sit in an ice bath for 15 min. The concentrations of the total protein were determined using a Modified BCA Protein Assay Kit (Sangon Biotech). Aliquots corresponding to 50 μg of each sample were analyzed using Western blotting (WB) analysis, which was carried out routinely as stated above (*potD* Mutant Identification). LPS group and PBS group were set as positive and negative control, respectively. The primary antibodies included p-Erk1/2, Erk1/2, p-p38-MAPK, p38-MAPK, p-JNK, JNK, p-p65(NF-κB), p65(NF-κB), TLR2, and TLR4. All antibodies were obtained from Santa Cruz Biotechnology Inc., Santa Cruz, CA, USA. The grayscale bands were developed by the ChemiDoc^TM^ XRS+ system with Image Lab^TM^ software (Bio-Rad). Band luminosity intensity was quantified using ImageJ software. IOD values were analyzed by AlphaEaseFC software, wherein GAPDH was used as internal reference gene.

### 2.13. Determination of Working Concentrations of Inhibitors Using MTT Assay

Prior to inhibiting cell-signaling pathways using inhibitor agents SB203580 (p38-MAPK inhibitor), U0126 (Erk1/2 inhibitor), SP600125 (JNK inhibitor) and PDTC (NF-κB inhibitor), the cytotoxicities of these signaling pathway inhibitors was evaluated in vitro using MTT colorimetric assay. 3-(4,5-Dimethylthiazol-2-yl)-2,5-diphenyltetrazolium bromide (MTT) was obtained from AMRESCO Inc. (Radnor, PA, USA). The macrophages were seeded in triplicate in 96-well plates at a density of 5000 cells/well and incubated at 37 °C, 5% CO_2_ overnight to allow the cells to attach to the plate. The medium was removed, and cells were re-incubated with complete DMEM/F-12 medium (Hyclone, Logan, UT, USA) for 24 h at 37 °C, supplemented with different concentrations of inhibitor agents (0, 1, 3, 5, 10, 20 μM each inhibitor per well). Afterwards, treated medium was removed and aliquots of 20 µL of MTT in 5 mg/mL concentration were added to each experimental well. The cocktail was incubated for another 4 h to allow MTT to be reduced to formazan. MTT was removed carefully and the intracellular purple formazan crystals were solubilized by DMSO. Plates were shaken for 10 min before reading them on a spectrophotometer (SmartspecTM Plus, Bio-Rad). The absorbance wavelength was recorded at 570 nm. Cytotoxicity was determined as inhibition ratio, which equals to [1 − (A − A_blank_)/(A_c_ − A_blank_)], in which A represents the A_570nm_ of experimental groups (inhibitor agent-treated); A_blank_ represents background wavelength in blank wells (no cell and no agent); A_c_ represents A_570nm_ in control groups, where cells were treated with MTT but no other inhibiting agent.

### 2.14. Effect of Inhibitor on the MAPK and NF-κB Signal Pathways

Raw 264.7 macrophages (1 × 10^6^ cells/mL) were cultured in 12-well cell culture plates. Cells were pretreated with the following specific inhibitors (10 µM each) in parallel for 30 min prior to PotD addition: SB203580 (p38 inhibitor), U0126 (Erk1/2 inhibitor), SP600125 (JNK inhibitor), PDTC (NF-κB inhibitor). Another DMSO (AMRESCO Inc., USA) group was set as a control (final concentration of 1 μL/mL). Inhibitors for studying signal pathways were obtained from MCE (10 mM each, Monmouth Junction, NJ, USA), dissolved in DMSO. Raw 264.7 were spun down and supernatant was collected for routine detection of cytokines IL-1β, IL-6, and TNF-α using ELISA kits. Results were analyzed to determine cytokine response levels and to confirm identity of cell-signaling pathways contributing to responses to rPotD exposure.

### 2.15. Statistical Analysis 

Comparisons of several independent test series were evaluated by analysis of two-way analysis of variance (ANOVA tests). Multiple comparisons between any two means of different groups were performed using least significant difference (LSD) method. Values were presented as mean values plus or minus the standard deviation from the mean (SD). A *p* value was generated for all variables, with *p* < 0.05 considered to be statistically significant (*).

## 3. Results

### 3.1. Purification of rPotD Protein, Endotoxin Elimination, and Polyclonal Antibody Preparation

rPotD protein was expressed as a fusion protein in its active form with both of its C-terminal and N-terminal 6 × His tags, facilitating the purification process using a Ni affinity chromatography in vitro. The products were dialyzed, ultra-filtrated and analyzed using 12% SDS-PAGE, while the *E. coli* BL21 strain harboring the empty pET-28a (+) vector alone served as a negative control. One band with the molecular mass of approximately 39 kDa, corresponding to rPotD protein plus its recombinant tag, was obtained as expected (Figure 1), indicating that the rPotD protein was successfully expressed in BL21(DE3) under the induction by 1 mM of IPTG for 5 h at 30 °C. After endotoxin removal, the concentration of endotoxin was less than 0.1 EU per mL in the purified rPotD. The purified rPotD was subsequently used to vaccinate mice and treat macrophages.

A week after booster immunization, blood samples from each mouse were collected by retro-orbital bleeding. After the serum was precipitated, a 96-well plate coated with purified rPotD protein was used to detect the specific antibody titers of mouse antiserum against rPotD via ELISA. Results showed that immunization with 0.1 mg of rPotD could sufficiently elicit a specific antibody titer up to nearly 1 × 10^6^ (Figure 2). The antiserum was adopted to determine the construction of *potD* gene mutant and complemented strains in later studies.

### 3.2. Recombinant PotD Protein Mediates a Cellular Immune Response

In addition to the finding that rPotD can actively elicit the humoral immune response, we further performed IF and FCM analyses to determine whether rPotD could trigger the upregulation of certain T cells in mice. The spleens of three mice in each group were treated with FFPE and used for routine IF analysis. The fluorescence staining showed that all CD molecules were located on the surface of spleen lymphocytes, forming circular fluorescence bodies (Figure 3B). IOD_R_ assay of normal spleen tissues revealed low-threshold immunofluorescence in white pulp area (Figure 3A). However, no conspicuously visible CD molecule could be observed in red pulp area. Closer inspection of the relative expression levels of immunocyte CD3^+^/CD4^+^/CD8^+^ revealed that all CD3^+^/CD4^+^/CD8^+^ were upregulated to almost twice that of the mock group upon inoculating rPotD+adjuvant in mice (*p* < 0.05). Clumping of T cells CD3^+^ and CD4^+^ clustered in white pulp area, while CD8^+^ was scattered in red pulp area, despite a low fluorescence intensity. However, mice immunized with pure adjuvant solely or injected with PBS only showed insignificant fluctuations in cell line levels, when compared with the mock group (*p* > 0.05) (Figure 3B).

In FCM assay, total T cells (mature T lymphocytes) and the T cell subsets T-helper cells (CD4^+^) and T suppressors/cytotoxic T-cells (CD8^+^) were quantitated in a representative peripheral blood sample of untreated or immunized mice (fluorescence-activated cell sorting, FACS, Appendix A). The distribution of CD8^+^ leukocyte infiltrates was somewhat different from the findings in IF assay. In two independent experiments, CD8^+^ respectively accounted for 14.18% and 14.61% of the whole cells (Appendix A), which were even slightly lower than those in the mock group, presumably because the different sensitivities in different detection methods and/or different samples (spleen tissue vs. peripheral blood), but the differences were not significant (*p* > 0.05). However, immunized mice-blood samples displayed magnification of CD3 and CD4 T lymphocytes (CD3^+^, CD4^+^) frequencies (*p* < 0.05). Relative CD3^+^ counts rose from 24.3% to a robust 33.76%/40.4% cells; CD4^+^ increased from 16.61% to 23.86%/29.08% cells (*p* < 0.05). On the other hand, CD4^+^/CD8^+^ increased from 0.931 (16.61/17.84) to 1.648 (23.86/14.48)/1.990 (29.08/14.61); this was presumably attributable to the immunization of rPotD. Significantly, both the CD3^+^ and CD4^+^ count and CD4^+^/CD8^+^ ratios were upregulated almost one and a half times that observed in the mock group, which was in accordance with the findings in the IF assay, where their relative fluorescence intensities (IOD_R_) nearly doubled in mice inoculated with rPotD+adjuvant. 

Overall, these observations corroborated the suggestion that rPotD can actively mediate a cellular immune response through stimulating the expressions of CD3^+^, CD4^+^ and elevating CD4^+^/CD8^+^ in mice. 

### 3.3. Lymphocyte Proliferation Assay and Detection of Cytokines

The lymphocyte proliferation assay set out with the aim of assessing the importance of possessing the level of rPotD immunogenicity and immune-protective efficacy. Splenocytes of mice two weeks after booster immunization were isolated and stimulated with rPotD protein and ConA. Levels of lymphocyte proliferation in each group were detected using an MTS method and the results shown as absorbance at A_490nm_. As shown in Figure 4, a significantly ascending proliferative lymphocyte immune response was observed in the mice immunized with rPotD+adjuvant (*p* < 0.05). Similarly, a strong lymphocyte cell proliferative response was detected when stimulated with ConA as a positive control (*p* < 0.05). However, compared with mock group, neither rPotD-alone group nor adjuvant-alone group elicited a significant proliferative splenocyte response when using rPotD, rather than ConA, as a stimulant. The proliferative splenocytes response may prove useful in blunting pathogenicity levels induced by *H. parasuis* infection.

Cytokines (CK) are signaling molecules that modulate inflammatory response components of the immune system during tissue damage and repair. Cytokines can be divided into interleukin (IL), interferon (IFN), and tumor necrosis factor (TNF) superfamily, colony stimulating factor (CSF), and chemokine and growth factor (CF and GF). Here, we preliminarily tested two interferons (IL-1β and 4) and an interferon (IFN-γ) to determine the immune response type ensuing rPotD stimulation. The levels of pro-inflammatory cytokines IL-1β and IFN-γ from the supernatants of splenocytes were stimulated with rPotD in all experimental groups (splenocytes from mice immunized with PBS/rPotD alone/adjuvant alone/rPotD+adjuvant) were significantly higher in the mice vaccinated with rPotD when compared with the mock groups (Figure 5A,C, *p* < 0.05). However, there was no increase in IL-4 levels in all experimental groups (Figure 5B, *p* > 0.05), indicating that immunization of mice with rPotD triggered a Th1-type immune response, rather than Th2-type.

### 3.4. rPotD-Vaccinated Mice Show Less Severe Pathological Signs in Multiple Tissues 

Given that rPotD can actively elevate lymphocyte proliferation and induce Th1-type immune response related to pro-inflammatory cytokines, we first evaluated the immunoprotective effect of rPotD in the mice model, prior to validating its immune protective effect in piglets. The rPotD+adjuvant group showed a mortality rate of 30% (3/10) upon challenge with *H. parasuis* SH0165, while the PBS/rPotD-alone/adjuvant-alone groups displayed rates of 100% mortality within 48 h (Figure 6A). Through monitoring, we found mice that survived in the rPotD+adjuvant presented less severe clinical symptoms than the mice in the PBS/rPotD-alone/adjuvant-alone groups. Most infected hosts in the latter group scored 5–7 within 24–48 h post-infection using the murine activity scores system, while most mice in rPotD+adjuvant group scored 1–4 or less within 48–72 h post-challenge; nearly all of them recovered fully, minus the dead hosts. No anomalies were observed in the mock group throughout the experiment. In passive immunization, all the mice receiving antisera elicited by PBS/rPotD alone/adjuvant alone died after the challenge with *H. parasuis* SH0165 within 36 h. One half of mice (5/10) immunized passively with antisera from rPotDmixed with adjuvant were protected from the lethal *H. parasuis* challenge (Figure 6B), and nearly all of the survivors recovered fully within 72 h. In the active immunization experiment, after the mice were humanely sacrificed, hearts, livers, spleens and lungs were harvested and treated with FFPE, and further utilized for H&E analysis to determine pathological lesion severity levels. As shown in Figure 7 (lungs) and Appendix A (hearts, livers, spleens), pathological lesions were present in all immunized groups. However, compared with PBS/rPotD-alone/adjuvant-alone groups, mice in rPotD+adjuvant group showed less severe pathological lesions. In the rPotD+adjuvant group, lung sections exhibited alveolar wall thickening accompanied by certain pulmonary lymphocytic infiltration; this group also displayed local alveolar dilation, thinned alveolar walls, and extrusion of surrounding alveoli, and a small amount of alveoli wall rupturing (Figure 7); spleen sections showed small areas of myocardial fiber necrosis, nuclear pyknosis, deep staining or dissolution, accompanied by a certain amount of lymphocytic infiltration in several pathology bearing regions (Appendix A). Other areas displayed no such pathological changes. By contrast, in the adjuvant-alone group, lung sections showed severe alveolar wall epithelial cell necrosis, nuclear pyknosis, and deep staining or fragmentation accompanied by a mass of lymphocytic infiltration; capillary congestion, large amounts of red blood cells in the lumen of the blood vessels, and hemorrhaging were also observed (Figure 7); spleen sections exhibited deeper level of myocardial fiber necrosis, nuclear dissolution, cytoplasmic loose staining, and significant myocardial fiber edema surrounding the tissue (Appendix A). Heart and liver sections displayed similar pathological signs (Appendix A). No anomalies were observed in the mock group.

### 3.5. rPotD+Adjuvant-Vaccinated Mice Showed Significantly Elevated Bacterial Clearance Ability

To assess the specific pathogen scavenging ability of PotD in the innate immune response in vivo, we subjected SPF BALB/c mice to a HLD of *H. parasuis* strain SH0165. Lung tissue homogenates were diluted serially and cultured on nalidixic acid-containing TSA++ overnight prior to counting the visible bacterial colonies (in a series of dil. 10^1^–10^7^). In the mock group, the number of visible bacteria ranged from 125–327 CFU/mL in active immunization, and 69–204 CFU/mL in passive immunization, severing as a background reference value. As expected, a statistically significant decrease in viable bacteria was detected in lung tissue homogenate from rPotD+adjuvant-immunized mice compared to those from all the other non-effective immune groups (Figure 8). Thus, we concluded that Hp–rPotD could trigger effective clearance of *H. parasuis* during acute pneumonia to prevent systemic infections. This may underscore the way in which rPotD+adjuvant-immunization could protect mice from certain dose of *H. parasuis* infection under lab conditions and may possess the potential to endow the host with the resistance against *H. parasuis* colonization in lungs in murine models.

### 3.6. rPotD Actively Elevates the Levels of Pro-Inflammatory Cytokines IL-1β, IL-6, and TNF-α

We have demonstrated using a mouse model that rPotD can effectively induce Th1-type immune responses in spleen cells. To analyze the function of PotD antigen in pathogenesis, pro-inflammatory cytokine responses were measured on Raw 264.7 macrophages. After macrophages were treated with rPotD protein, secreted cytokines IL-1β, and IL-6, and TNF-α in the supernatant of cell culture were determined using ELISA kits. After treatment with rPotD, the secretion level of all three cytokines in the supernatant of cell culture was significantly upregulated compared with the negative control (*p* < 0.05). Additionally, 30 μg/mL of rPotD triggered a statistically significant increase in all three cytokines than 15 μg/mL of rPotD, showing a strong correlation between higher rPotD concentration and increased cytokine induction levels. LPS (200 ng/mL) served as a positive control, which promoted the upregulation of the production of all these cytokines, while low secretion levels of IL-1β, and IL-6, TNF-α were obtained in the negative control (treated with PBS) (Figure 9).

In transcriptional levels, the relative quantification of 2(−ΔΔC(T))-based qRT-PCR method was performed to test transcriptional levels of all three cytokines. GAPDH was utilized as internal reference gene. The mRNA levels of IL-1β, IL-6, and TNF-α were significantly increased (Appendix A, *p* < 0.05) compared with negative controls, indicating activation and pro-inflammatory activity of macrophages. Likewise, 30 μg/mL of rPotD displayed statistically significant increased transcriptional levels in all three cytokines than 15 μg/mL of rPotD, which was in correlation with ELISA results, showing that rPotD can actively promote the Th1-type pro-inflammatory cytokines activation of macrophages at both protein and mRNA levels. 

### 3.7. TLR4, Rather Than TLR2, Participates in Regulatory Role on Expression of Pro-Inflammatory Cytokines in Macrophages

Toll-like receptors (TLR) 2 and 4 were separately or collectively blocked using anti-TLR2 and anti-TLR4 monoclonal antibodies in an effort to prove that rPotD upregulated cytokine production through binding with macrophage cell surface recognition receptors. ELISA results demonstrated that production of the three pro-inflammatory cytokines increased significantly after TLR4 was blocked compared with the rPotD group (Figure 10), especially IL-6 and TNF-α, which declined from 95.68 ± 2.34 and 112.73 ± 1.74 pg/mL to 46.36 ± 0.28 and 67.68 ± 7.68 pg/mL, respectively. The declines were nearly two-fold. However, in the anti-TLR2 or anti-TLR2/4 group, secreted levels of all of the three pro-inflammatory cytokines had no significant changes (*p* > 0.05). Taken together, these results indicated that TLR4, rather than TLR2, mediated PotD-induced expression of pro-inflammatory cytokines in macrophages in *H. parasuis* infection.

### 3.8. Determination of Working Concentrations of Inhibitors Using MTT Assay

An MTT assay was carried out to determine the cytotoxicities of signaling pathway inhibitors SB203580 (p38-MAPK inhibitor), U0126 (Erk1/2 inhibitor), SP600125 (JNK inhibitor), and PDTC (NF-κB inhibitor), prior to being applied to block their corresponding signaling pathways. According to manufacturing instructions, the recommended concentration for SB203580/U0126/SP600125/PDTC was 1–50 μM/1–20 μM/1–100 μM/10–100 μM, respectively. However, in this study, we found 10 μM of each signaling pathway inhibitor induced a slight inhibiting effect on cell line Raw 264.7, while 20 μM of each inhibitor, especially the SB203580 and SP600125, demonstrated significant higher levels of inhibitory effects on cell growth (inhibitory rate > 10%) (Figure 11). The Raw 264.7-inhibitory rates for SB203580/U0126/SP600125/PDTC were 26.65%/12.60%/22.97%/14.37%, respectively. Thus, in later studies, we used 10 μM each as the optimized working concentration, which was efficacious enough that this concentration was much higher than their corresponding IC_50_ for signaling pathway inhibition according to the product specifications, and 10 μM of these inhibitors had negligible cytotoxic effects to cell line Raw 264.7.

### 3.9. Production of Pro-Inflammatory Cytokines Mainly through JNK–MAPK and NF–κB Signal Pathways

For further investigation of the signal pathways for PotD engaged in boosting pro-inflammatory cytokines, Western blotting was carried out to determine three MAPK pathways, including p38, ERK, JNK signal molecules, and another RelA(p65) signal molecule belonging to the NF-κB pathway. As outlined in Figure 12, only the phosphorylation of JNK–MAPK in Raw 264.7 cells induced by PotD stimulation was augmented significantly, while the expression levels of the other signal molecules fluctuated within a statistically insignificant range (*p* > 0.05). The raw data of western blotting used in this study in Appendix A.

Expression of TLR2 and TLR4 in macrophages was further assayed in this test (Figure 12). After treatment with PotD, TLR4 was significantly upregulated in Raw 264.7 compared with the negative control, while the expression of TLR2 was relatively constant. The WB results for TLR2 and 4 were similar to those in the TLR-blocked experiments as above, which confirmed the observation that TLR4, rather than TLR2, participates in regulatory role on expression of pro-inflammatory cytokines in Raw 264.7 macrophages.

In a follow-up study, we aimed to characterize the details of the stimulation of signal pathway stimulation by rPotD, inhibitors SB203580 (p38–MAPK inhibitor), U0126 (Erk1/2–MAPK inhibitor), SP600125 (JNK–MAPK inhibitor), and PDTC (NF–κB inhibitor) were applied to block their corresponding signal pathway in Raw 264.7, with their supernatants being collected for detection of cytokines IL–1β, IL–6, and TNF–α via ELISA. Figure 13 plots the secretion levels of these three cytokines under different treatments, wherein a DMSO group was set as a control in parallel. Results showed that all these cytokines were downregulated significantly when the JNK–MAPK signal pathway was blocked, which coincided with our finding in the WB assay. However unexpected and noticeable, in ELISA, these cytokines were also downregulated when p65 (NF–κB) was blocked, especially the secretion levels of IL–1 and –6. We postulated that ELISA is used to detect the functional output of the signal pathways and WB is used to detect the protein expression level change. Thus, we concluded that the NF-κB inhibitor (PDTC) also induced a certain degree of reduction. These results suggest that the PotD-induced cytokine production likely primarily depends on the phosphorylation of JNK–MAPK signal molecule and, to some extent, harnesses the NF–κB signal pathway as an alternative two-tiered circuit.

## 4. Discussion

*Haemophilus parasuis*, or *Glaesserella parasuis* (*G. parasuis*) according to the new taxonomic reclassification, is normally a benign swine commensal in upper respiratory tract but may cause severe vascular lesions and multiorgan dysfunction, eliciting serious economic losses annually worldwide [46,47]. Commercial bacterins are often used to vaccinate swine against *H. parasuis*, but the deficiency in crossreactivity makes sole-serotype inactivated vaccine an ineffective means of protection against this disease [48].

Polyamines, such as putrescine, cadaverine, spermidine and spermine, constitute a group of small, ubiquitous aliphatic, polycationic molecules which are present in all cells [49]. They play a myriad of cellular physiological roles in both eucaryotes, such as in mung bean sprouts and cancer cells, and in prokaryotes, such as Ebola virus and *E. coli* [50,51,52,53]. In numerous landmark studies, the most important biological functions of polyamines have been well elucidated. However, only a small proportion of them have focused on the exact role of polyamines in microorganisms. A lipoprotein signal peptide was detected in the HpPotD with a cleavage site between position A22 and N23, and TMHMM2.0 server demonstrated that HpPotD has no typical transmembrane helix, suggesting that the protein is transported as a precursor protein through the cell membrane, lipid modified and retained at the inner membrane. This implied that this protein was more likely to be hydrophilic, which is consistent with the GRAVY value (−0.331) of HpPotD calculated by ExPASy ProtParam tool. Further studies concerning polyamine uptake and physiological functions in *H. parasuis* are needed. 

Previous research findings confirmed the effectiveness of PotD protein in generating both vigorous and highly specific mucosal and systemic immune responses, and in reducing nasopharyngeal carriage and invasive disease caused by multiple pneumococcal serotypes [28]. Another recent study confirmed that chimeric proteins based on surface antigens PspA in fusion with PotD (rPspA–PotD fusion) were able to elicit strong immune responses and to protect against invasive pneumococcal infection, thus could reduce nasopharyngeal colonization in mice [54]. In our recent study, we found PotD can actively mediate both the humoral immune response and cellular immune response. Prior to determining whether SC1401 PotD can protect mouse from infection of the highly virulent *H. parasuis* strain SH0165 (serotype 5), we evaluated several immune indices. Through IF and FCM analysis, we found PotD could stimulate the expression levels of CD3^+^, CD4^+^ and elevate CD4^+^/CD8^+^ in mice when compared with mock groups. Lymphocyte proliferation results visually showed that mice can positively respond to PotD-stimulation. ELISA screens demonstrated that cytokines IL–-1 and IFN–γ were boosted significantly after the primary spleen cells were exposed to PotD. However, the Th2-type immune response representative IL–4 was not affected. The observation of the effectively immunoprotective effect of PotD on the mouse model prompted us to further investigate the specific mechanisms of induction of immunity. Prior studies have noted the importance of TLRs in microbial identification and intrusion. Bacterial surface molecules are core factors in the microorganism–host crosstalk, a well-studied pattern by which TLRs recognize and bind with PAMPs/MAMPs [55]. Bacteria, viruses, fungi, and parasites can be recognized by TLR2. TLR4, with an adaptor molecule MyD88, is essential in the recognition of and response to antigenic components of multiple Gram-negative bacteria [56]. At present, in pro-inflammatory cytokines analysis and signal pathway investigation, we found TLR4 and not TLR2, was involved in positive control of cytokine production by PotD in *H. parasuis*, which was different from a previous finding for GALT in *A. pleuropneumoniae*, in which TLR2 played a contrary role against TLR4. The blocking of TLR2 significantly augmented expression levels of relative downstream pro-inflammatory cytokines induced by GALT in *A. pleuropneumoniae* [43]. However, the secretion levels of cytokines were still much higher than control through blocking TLR-4. We speculate that TLRs, especially TLR4 in this case, are not the only PRRs in the process of antigen recognition. More effort is needed to elaborate such a phenomenon. Further investigation in signal pathways confirmed that both JNK–MAPK and NF–κB play primary roles in the pro-inflammatory response induced by PotD stimulation in macrophages. Trigging expression of pro-inflammatory cytokines is a central phenomenon in mediating a variety of immune responses. Our research sheds light on the positive effect of PotD in aggravating inflammatory responses of macrophages in mice.

## 5. Conclusions

Taken together, one of the primary findings to emerge from this study is that the antibodies to the conserved polyamine transporter PotD confer effective protection against invasive GD within murine models, which hopefully lays a reasonable foundation for further verification of its immunoprotection in piglets. We also conclude that Hp–rPotD can actively elevate the secretion of pro-inflammatory cytokines IL–1, IL–6, and TNF–α of macrophages via JNK–MAPK and NF–κB signal pathways through TLR4, rather than TLR2. Our findings are hopefully expected to enrich the knowledge of developing improved strategies for the control of *H. parasuis* infection in the future.

## Figures and Tables

**Figure 1 vaccines-07-00216-f001:**
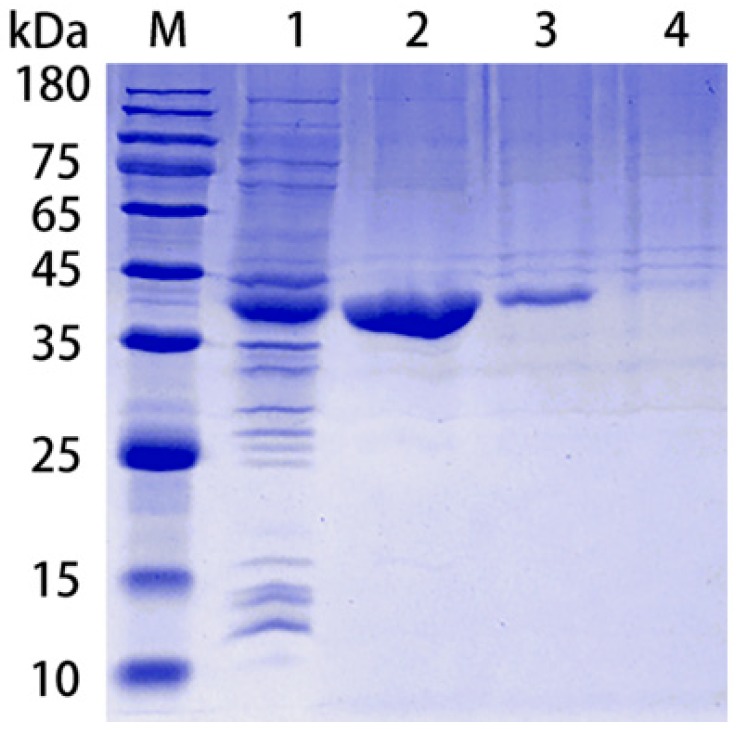
Purification of rPotD protein. Lane 1: Ultrasonic-lysed crude protein of pET-*potD*-BL21(DE3); lane 2: purified rPotD protein (dialyzed, ultra-filtrated); lane 3: purified rPotD protein (without being ultra-filtrated); lane 4: low concentration of elution buffer; M: protein molecular mass standard (Novoprotein).

**Figure 2 vaccines-07-00216-f002:**
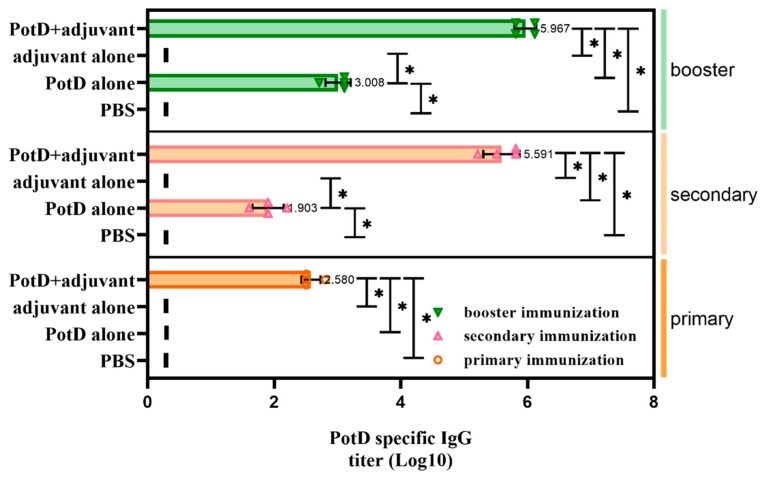
Determination of specific IgG titers in serum samples collected from each mouse after receiving three rounds of immunization. IgG titers were determined using indirect ELISA coated with purified recombinant PotD protein. The titer was defined as the absorbance of the highest dilution at 450 nm which was at least 0.1 above that of the background wells in which serum samples were replaced with PBS. Short vertical lines indicate value below detection limit. These experiments were repeated four times. Each error bar represents one standard deviation from the mean.

**Figure 3 vaccines-07-00216-f003:**
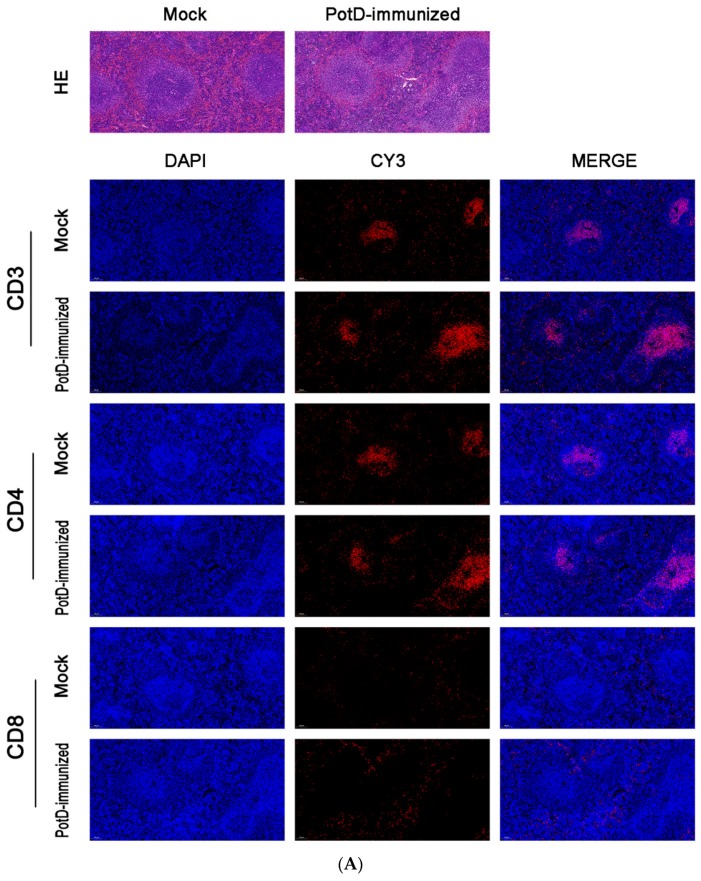
Immunofluorescent (IF) assay for the level of CD positive T-cells in spleen tissues. Spleen tissues were harvested from mice in different groups after 7 dpi, and used for FFPE treatment, pathological slice preparation, and IF assay for the levels of immunocyte CD3^+^/CD4^+^/CD8^+^. (**A**) IF analysis and H&E reference points (100×). (**B**) IF analysis (725×) and statistical analysis of IOD of T lymphocyte CD3^+^/CD4^+^/CD8^+^. Relative values of IOD_R_ were calculated by the ratio of the IOD of CY3-fluorescence intensity (IOD_C_) and DAPI (IOD_D_).

**Figure 4 vaccines-07-00216-f004:**
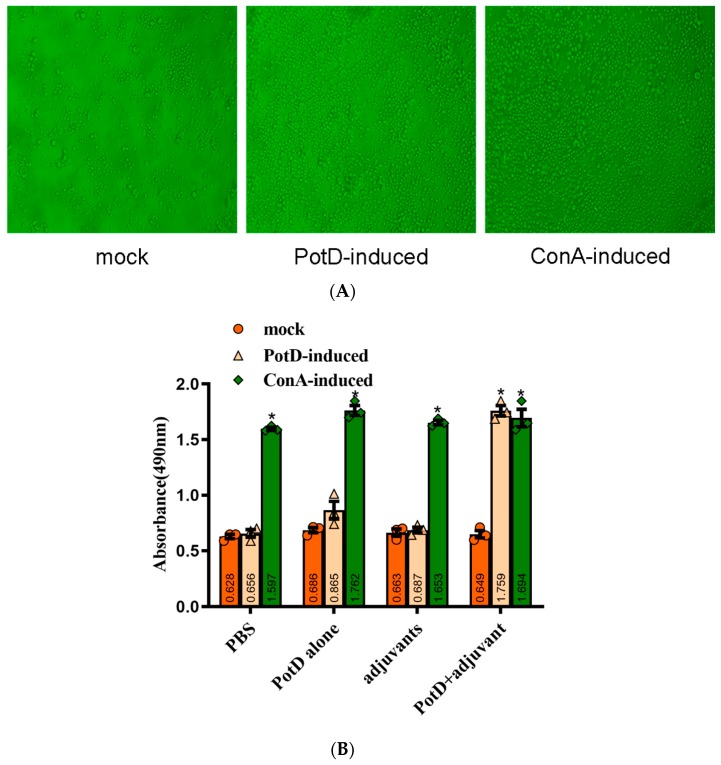
Lymphocyte Proliferation Assay. Two weeks after the booster immunization, the splenocytes (1 × 10^6^ cells/well) from immunized groups and mock group were cultured with 10 μg/well of proteins and ConA (positive control) for 72 h at 37 °C with 5% CO_2_. Levels of lymphocyte proliferation in each group were detected using an MTS method and the absorbance at 490 nm (A_490nm_) was measured using a Varioskan Flash (Thermo Scientific). (**A**) Microexamination of spleen lymphocyte proliferation in the mice immunized with rPotD+adjuvant. (**B**) Spleen lymphocyte proliferation analysis. These experiments were repeated three times. Each error bar represents one standard deviation from the mean.

**Figure 5 vaccines-07-00216-f005:**
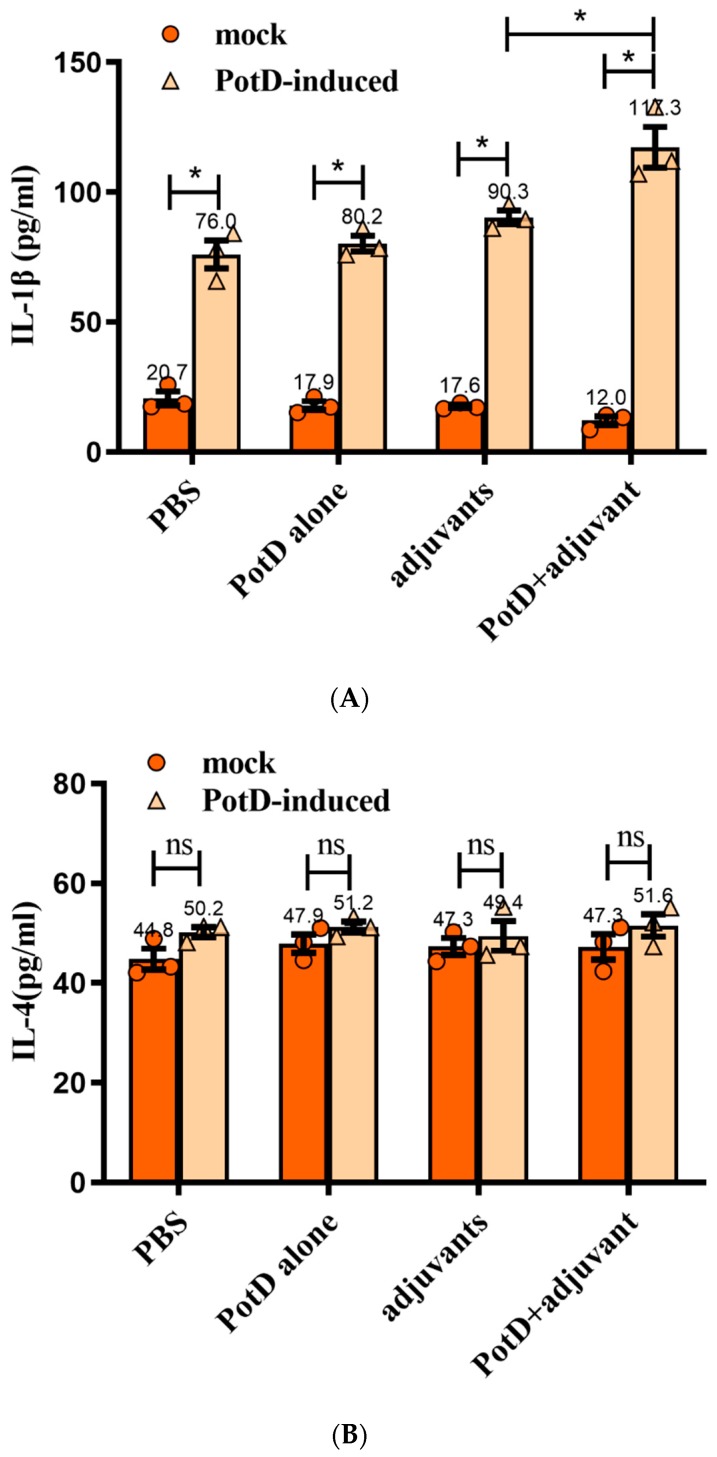
The concentrations of (**A**) IL-1, (**B**) IL-4, and (**C**) IFN-γ in cultured primary spleen cells of mice. Production in cultured spleen cells of mice injected with proteins plus MONTANIDE™ GEL 01 adjuvant or the adjuvant control (negative control). Results are expressed in picograms per milliliter. The asterisk ***** indicates significance at a *p* < 0.05 level. These experiments were repeated three times. Each error bar represents one standard deviation from the mean.

**Figure 6 vaccines-07-00216-f006:**
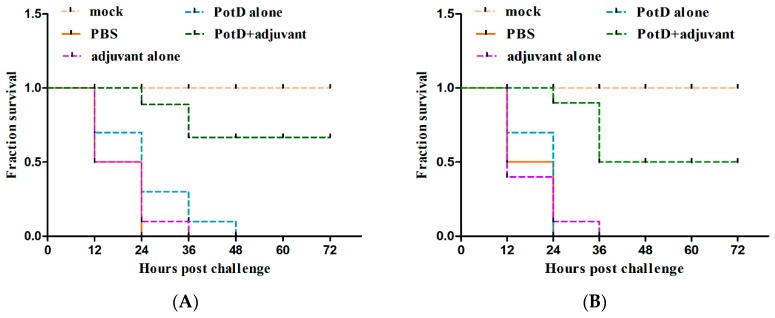
The survival rate of mice challenged with a lethal dose (1.4 × 10^9^ CFU/mouse) of *H. parasuis* SH0165. (**A**) In active immunization, PBS/rPotD-alone/adjuvant-alone groups triggered 100% (10/10) death in mice model within 48 h post-injection, while rPotD+adjuvant group engendered only 30% (3/10) of death within a 72 h-period observation. (**B**) In passive immunization, all the mice receiving antisera elicited by PBS/ rPotD alone/adjuvant alone died after the challenge with *H. parasuis* SH0165 within 36 h. One half of mice (5/10) immunized passively with antisera from rPotD mixed with adjuvant were protected from the lethal *H. parasuis* challenge. Mock group indicates mice that were neither vaccinated nor challenged with SH0165, while PBS group indicates mice receiving both PBS inoculation and bacteria injection. All assays were performed at least two times independently. Survival data were recorded every 12 h for a period of 72 h and plotted using GraphPad Prism v. 8.0.2.

**Figure 7 vaccines-07-00216-f007:**
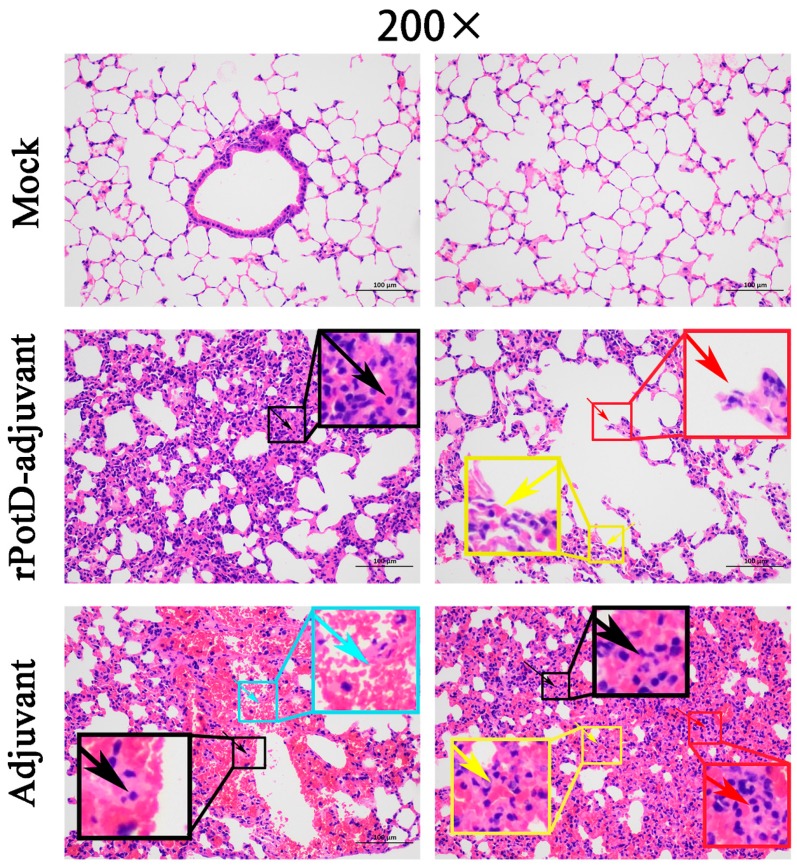
Histopathologic analysis of mice lungs (200×). Lung tissues were harvested from mice in different groups after 72 h dpi, and used for histological examination (H&E) staining and histopathological assay. Mock group (Mock): no visible pathological damage. rPotD-adjuvant group: thickening alveolar walls and inflammatory cells infiltration (black arrow); expanded alveoli, thinner alveolar walls (yellow arrow); broken alveolar walls (red arrow). Adjuvant-alone group: necrosis, karyopycnosis and hyperchromatism and fragmentation of the epithelial cells in the alveolar walls (black arrow); inflammatory cells infiltration (red arrow); dilated capillaries congested with red blood cells (yellow arrow); local hemorrhage (blue arrow).

**Figure 8 vaccines-07-00216-f008:**
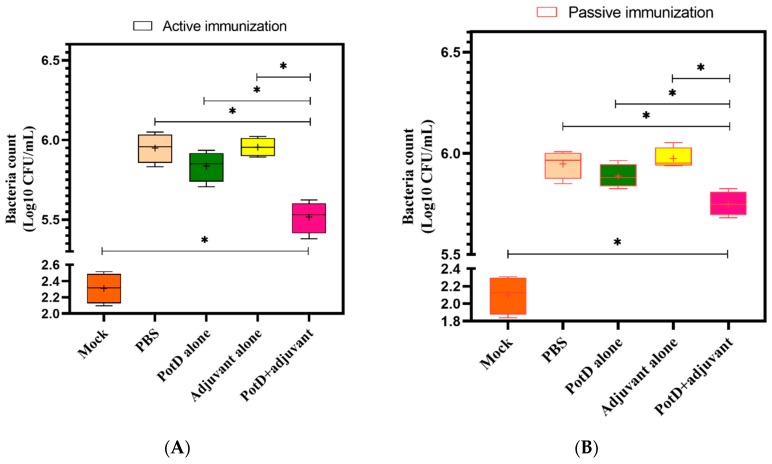
Viable nalidixic acid-resistant colonies were counted for *H. parasuis* isolated from PotD-immunized (**A**) or non-treated (**B**) SPF mice at 12 h postinjection. One hundred microliters per diluted lung tissue homogenate were plated on TSA++ supplemented with 200 µL nalidixic acid (100 mg/mL). Plates were incubated at 37 °C overnight, and visible bacterial colonies were counted manually. Graph represents mean ± SD for 4 mice/group for each treatment (***** indicates a significant difference (*p* < 0.05) between groups of immune treatment using paired *t*-test, and plus sign indicates mean value in each group.).

**Figure 9 vaccines-07-00216-f009:**
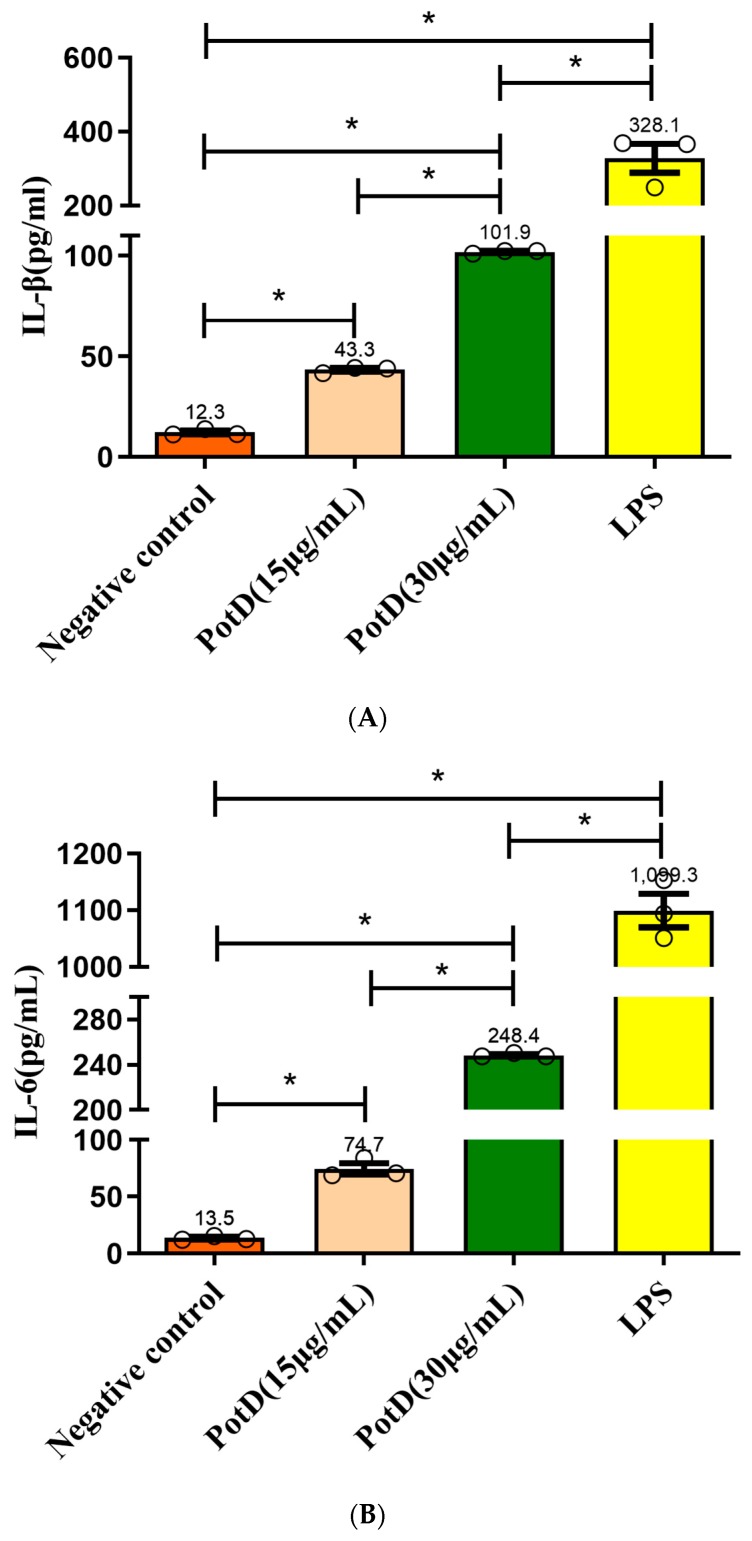
ELISA detection of the levels of pro-inflammatory cytokines induced by rPotD. Induction of cytokines in Raw 264.7 macrophages by rPotD stimulation. Raw 264.7 macrophages were incubated with 15 and 30 µg/mL rPotD, and 200 ng/mL LPS (positive control) for 12 h, as well as single culture media (negative control). Protein levels of (**A**) IL-1β, (**B**) IL-6, and (**C**) TNF-α in the culture supernatants were determined by ELISA. All assays were performed at least three times independently. Each error bar represents one standard deviation from the mean.

**Figure 10 vaccines-07-00216-f010:**
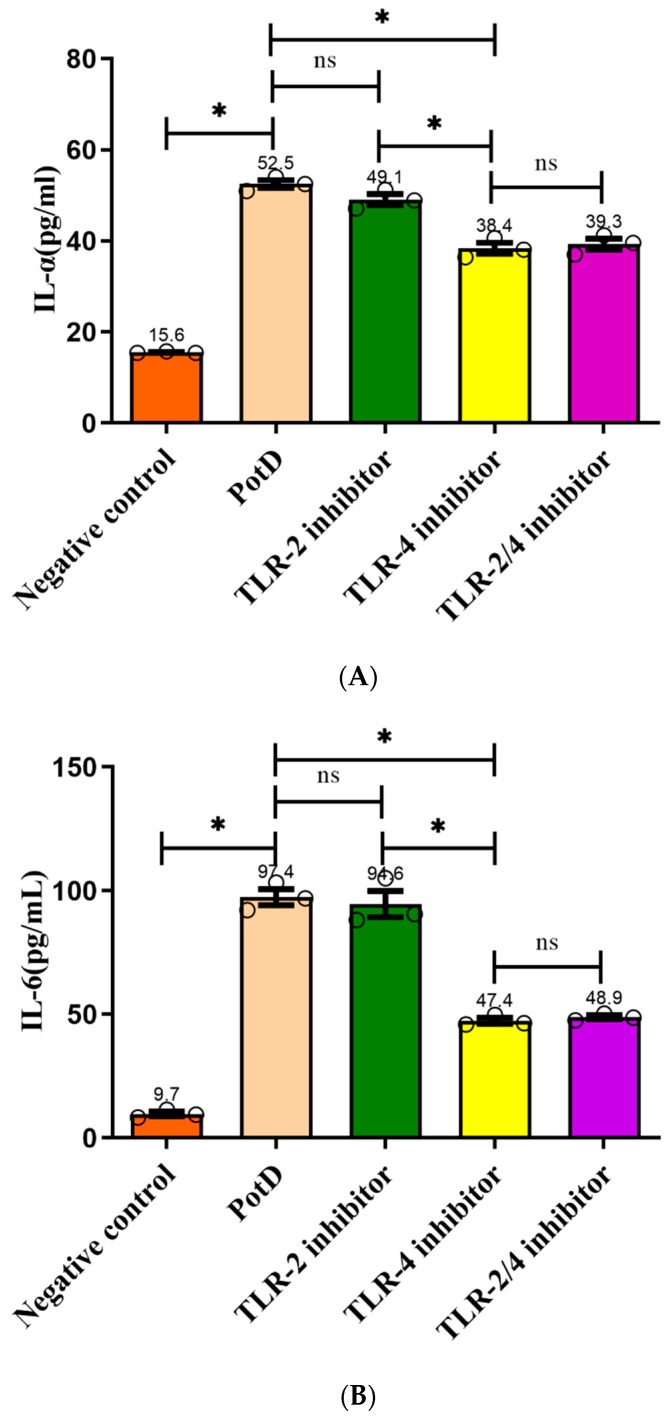
Analysis of the recognition receptor of PotD. Anti-TLR2 and anti-TLR4 monoclonal antibodies were used to block their corresponding Toll-like receptor or both TLR-2/4. Protein levels of (**A**) IL-1β, (**B**) IL-6, and (**C**) TNF-α in the culture supernatants were determined by ELISA. These experiments were repeated 3 times. Each error bar represents one standard deviation from the mean.

**Figure 11 vaccines-07-00216-f011:**
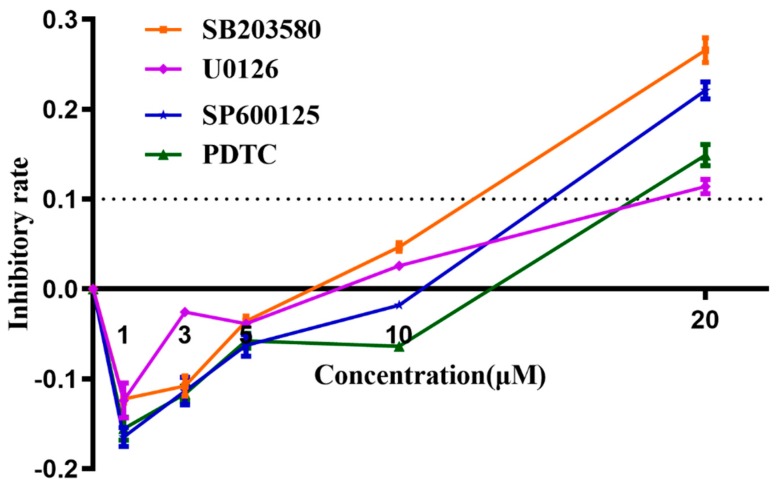
MTT Analysis of cytotoxicities of signaling pathway inhibitors (inhibitory rate curve). Cytotoxicities of signaling pathway inhibitors SB203580 (p38-MAPK inhibitor), U0126 (Erk1/2 inhibitor), SP600125 (JNK inhibitor), and PDTC (NF-κB inhibitor) were determined as an inhibition ratio, which is equal to (1 − (A − A_blank_)/(A_c_ − A_blank_)), in which A represents the A_570nm_ of experimental groups (inhibitor agent-treated); A_blank_ represents background wavelength in blank wells (no cell and no agent); A_c_ represents A_570nm_ in control groups, where cells were only treated with MTT agent, but not any inhibitor agent. These experiments were repeated 3 times. Each error bar represents one standard deviation from the mean.

**Figure 12 vaccines-07-00216-f012:**
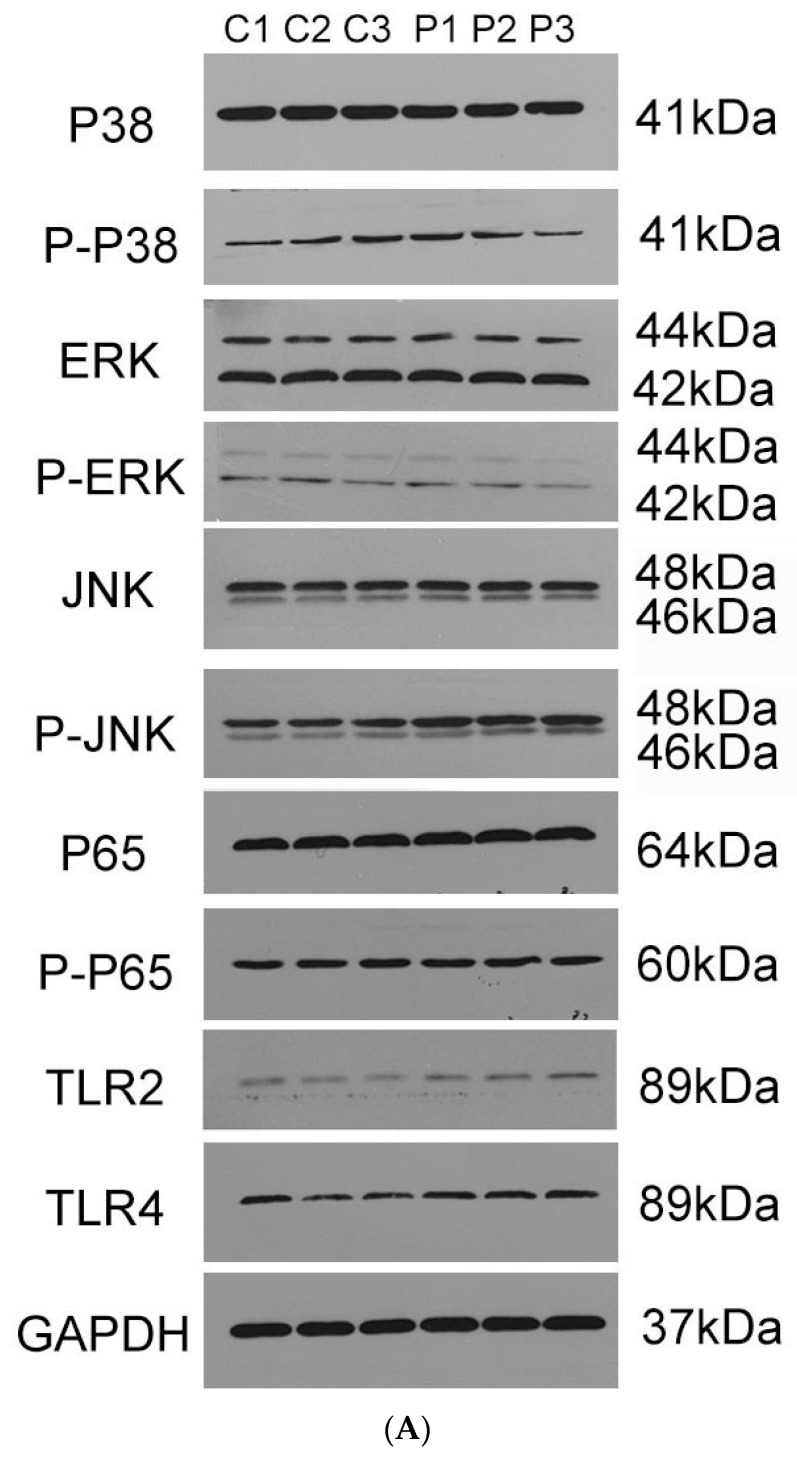
Western Blotting (WB) detection of the signaling molecules involved in cytokine secretion. Raw 264.7 macrophage was treated with 15 μg/mL of purified rPotD for 12 h, and used for detection of signaling molecule in MAPK and NK–κB signaling pathways. (**A**) Western blot analysis. (**B**) gray intensity. (**C**) Ratios between phosphorylated activated signals. C1–C3 represent three parallel mock groups, while P1–P3 represent three parallel Hp–rPotD challenged groups. These experiments were repeated 3 times. Each error bar represents one standard deviation from the mean.

**Figure 13 vaccines-07-00216-f013:**
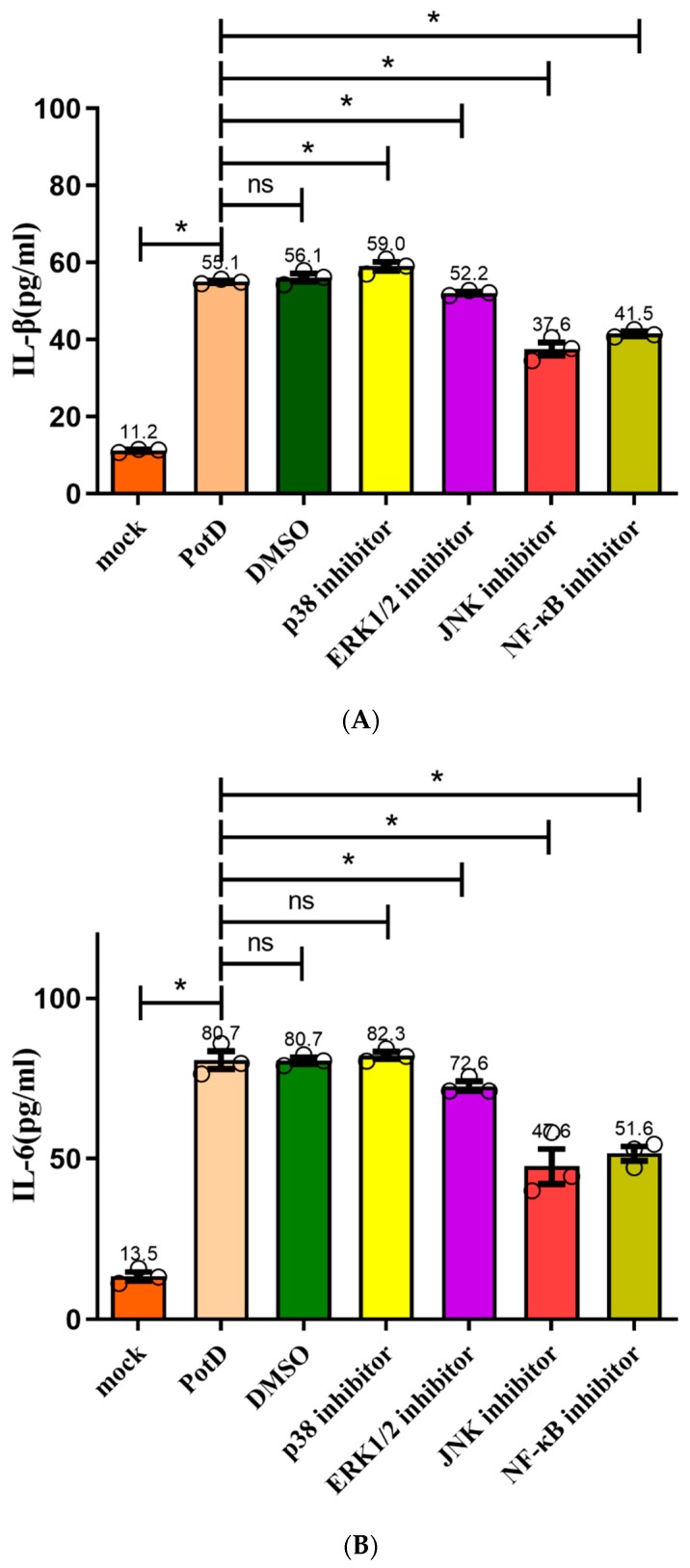
Effects of inhibitors on the pro-inflammatory cytokine secretion induced by PotD based on ELISA. Signaling pathway inhibitors SB203580 (p38–MAPK inhibitor), U0126 (Erk1/2 inhibitor), SP600125 (JNK inhibitor), and PDTC (NF–κB inhibitor) were used to block their corresponding MAPK and NF–κB signal pathways. Protein levels of IL–1β (**A**), IL–6 (**B**), TNF–α (**C**) in the culture supernatants were determined by ELISA. These experiments were repeated 3 times. Each error bar represents one standard deviation from the mean.

**Table 1 vaccines-07-00216-t001:** Primers used in this study.

Primers	Primer Sequences (5′→3′)	Products (bp)
**rPotD expression primers**		
P1 (*potD*-pET-F)	cagcaaatgggtcgc**ggatcc**AATGACACTGTACATCTTTATACTT	1015
P2 (*potD*-pET-R)	ctcgagtgcggccgc**aagctt**GCTTCGCAGCTTTTAACTCTTG	
**qPCR primers**		
P3(GAPDH-F)	GTGTTCCTACCCCCAATGTG	189
P4(GAPDH-R)	CATCGAAGGTGGAAGAGTGG	
P5(IL-1β-F)	GGGCCTCAAAGGAAAGAATC	183
P6(IL-1β-R)	TACCAGTTGGGGAACTCTGC	
P7(IL-6-F)	GGGACTGATGCTGGTGACAA	147
P8(IL-6-R)	TCCACGATTTCCCAGAGAACA	
P9(TNF-α-F)	CGTCAGCCGATTTGCTATCT	184
P10(TNF-α-F)	CTTGGGCAGATTGACCTCAG	

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
