# Peer review of "Polyamine Transport Protein PotD Protects Mice against Haemophilus parasuis and Elevates the Secretion of Pro-Inflammatory Cytokines of Macrophage via JNK–MAPK and NF–κB Signal Pathways through TLR4"

_vaccines, 2019, doi:10.3390/vaccines7040216_

Round 1
Reviewer 1 Report
Dai et al. investigated the potential use of the PotD recombinant protein from Haemophilus parasuis SC1401 (serotype 11) as a vaccine. They used mice and cultured cells in their study with the goal of providing immunoprotection to piglets in the future. This is an extensive analysis of PotD with some interesting results. Overall, the manuscript is well-written.
Minor points:
- Abstract should mention the use of mice.
- Bar plots should be plotted according to https://www.nature.com/articles/s41551-017-0079
- Error bars should be explained, i.e., standard deviation or standard error for how many biological replicates.
- Raw experimental data points should be made available as a resource to the community (e.g. via figshare).
- Abbreviations such as H&E should be defined.
- Page 16, Line 476, ‘positive correlation’ is misleading, should be replaced with other words such as ‘higher’, ‘stronger’, ‘increased’ etc.
- Fig 10A. C1, C2, C3, P1, P2, and P3 should be explained. Are C and P belong to mock and PotD, respectively?
Reviewer 2 Report
Dai and colleagues present a well written article intended at investigating the PotD protein utility in vaccination. The argument is of interest for pig farming, but authors used mice to test their hypothesis leaving verification in pig for further studies. The experiments are in general well designed but some raised perplexities:
Major points
Passive immunization data and methods are not shown but are essential for author’s conclusion that antibodies against PotD confer protection against infection. Authors should present those data and relative methods. Anti PotD antibody titers induced by PBS/PotD alone/adjuvants or PotD+adjuvants immunization should be reported. The authors challenged the mice with Haemophilus (Glaesserella) parasuis intraperitoneally. They get a systemic disease (sepsis?). To complement data in active and passive immunization protocols a quantification of bacteremia/bacterial clearance in the different experimental groups could be of use.Minor points
Data in Fig.2 indicates that upon PotD immunization the membrane levels of CD3, CD4 and CD8 goes up. This is somehow strange, I would expect that the number of positive lymphocytes goes up. Authors choose to measure % of CD3, CD4 and CD8 positive cells in peripheral blood samples (Fig. S1) and not in spleens and there CD8+ stay down. In Fig.3 A some photographs of cells are shown upon PotD or ConA treatment but is not immediately clear to wich experimental group they refers (PBS/PotD alone/adjuvants or PotD+adjuvants) In Fig.5 mock are 100% surviving probably they are not infected while PBS are infected but this is not immediately clear from the legend. In cultured spleen cells the authors measured IL-1B, IL-4 and INFg, why not IL-17? In Fig. 10 a positive control would be precious.
Final recommendation: reconsider after revision
Reviewer 3 Report
The manuscript entitled “Polyamine Transport Protein PotD Protects mice against Haemophilus parasuis and Elevates the Secretion of Pro-inflammatory Cytokines of Macrophage via JNK MAPK and NF-κB Signal Pathways through TLR4” by Ke Dai and co-workers, investigated the downstream mechanisms of how PotD might protects mice against H. parasuis through the immune systems. The key findings are Hp-rPotD can actively elevate the secretion of pro-inflammatory cytokines IL-1, IL-6 and TNF-α of macrophage via JNK MAPK and NF-κB signal pathways through TLR4, rather than TLR2. The authors did a great job to conduct all necessary experiments in an extensive way to support their conclusions. I do have some concerns about the manuscript:
1). The authors used the monoclonal antibodies to block the function of TLR-2 and TLR-4, respectively. And, the results suggest only blocking TLR-4 can bring down the secretion level of cytokines, but not TLR-2. However, the secretion level of cytokines is still much higher than control by blocking TLR-4. The authors will need to try blocking both TLR-2/4, to functionally analysis if TLR-2/4 functional redundant in this case.
2). The results present in Fig 3 and 4, which aims to conclude that PotD can increase the lymphocyte proliferation and also increase the cytokine secretion are confusing. Because the data can also be interpreted the other way, which is the adjuvants treatment increase both lymphocyte proliferation and cytokine, since treat alone with PotD and adjuvants have the same effect. How the authors will explain these results, and maybe add other controls.
3). The authors used both WB and ELISA assay to demonstrate that both JNK-MAPK and NF-κB are important for the secretion of cytokines. However, the author reasoned NF- κB might play a weaker role than JNK-MAPK, since the WB and ELISA assay gives different results. ELISA is used to detect the functional output of the signal pathways, and WB is used to detect the protein expression level change. They do not have to be correlated.
4). On page 4, line 157. The authors need to annotate “dpi”.
5). On page 5, line171. “Briefly, 100 μL portions of heparin anticoagulant peripheral blood cells were….”, delete “were”.
6). On page 8, line 312. Although the statistic annotation is arbitrary, people usually use one asterisk (*) for P < 0.05.
7). On page 24, line 616. Change “resent” to “recent”.
Round 2
Reviewer 2 Report
Authors made a lot of efforts to address my criticism convincing me of the soundness of their work.
I do think this article has sufficient merit to be published in Vaccines.